# A New Role for Old Friends: Effects of Helminth Infections on Vaccine Efficacy

**DOI:** 10.3390/pathogens11101163

**Published:** 2022-10-08

**Authors:** Feifan Zhu, Wenyi Liu, Tong Liu, Linpeng Shi, Wenwen Zheng, Fei Guan, Jiahui Lei

**Affiliations:** Department of Pathogen Biology, School of Basic Medicine, Tongji Medical College, Huazhong University of Science and Technology, Wuhan 430030, China

**Keywords:** helminth infection, vaccine, efficacy, immunological mechanism, anthelmintic treatment, SARS-CoV-2

## Abstract

Vaccines are one of the most successful medical inventions to enable the eradication or control of common and fatal diseases. Environmental exposure of hosts, including helminth infections, plays an important role in immune responses to vaccines. Given that helminth infections are among the most common infectious diseases in the world, evaluating vaccine efficiency in helminth-infected populations may provide critical information for selecting optimal vaccination programs. Here, we reviewed the effects of helminth infections on vaccination and its underlying immunological mechanisms, based on findings from human studies and animal models. Moreover, the potential influence of helminth infections on SARS-CoV-2 vaccine was also discussed. Based on these findings, there is an urgent need for anthelmintic treatments to eliminate helminth suppressive impacts on vaccination effectiveness during implementing mass vaccination in parasite endemic areas.

## 1. Introduction

Vaccines are one of the most successful medical inventions known to date, capable of the eradication or control of common and fatal diseases. The proper term “vaccination” dates back to 1796 by the British scientist Edward Jenner (1749–1823) [1]. He proposed that vaccination with cowpox could prevent smallpox, which embarks on the wide application of vaccination to prevent and control other infectious diseases. Effective vaccination depends on the development of strong, appropriate, and durable immune responses in the recipients. Although universal vaccination has significantly reduced mortality from various infectious diseases, vaccination efficacy varies at both individual and group levels [2,3]. It has been estimated that up to 77 million children vaccinated against tuberculosis, 18 million children vaccinated against poliomyelitis, 19 million children vaccinated against measles, and 10 million children vaccinated against pneumococcal and pertussis are still infected after vaccination [4]. The factors that cause inconsistency in vaccination efficacy and vaccine-specific immune responses are complex and diverse. In addition to the factors regarding vaccines themselves, such as type, adjuvant, and dose, environmental exposure of hosts plays an important role in host responses to vaccines [5,6].

Epidemiological surveys have found that mass vaccination is sometimes less effective than expected, especially in developing countries [7,8].The century-old Bacille Calmette-Guérin (BCG) and measles vaccines are reported to be less effective in Africa and Southeast Asia than in developed countries [9,10]. The failure of vaccination to achieve the desired protective effect is not only related to technical reasons, such as vaccine quality and vaccination method, but also related to environmental exposure, one of which is the prevalence of helminth infections in many regions around the world [11,12,13,14]. The World Health Organization (WHO) estimates that helminth infections affect more than a quarter of the population, and more than 880 million children require treatment for these parasite diseases, including approximately 350 million in the African region [15]. Studies with animal models have demonstrated that helminth-induced immunomodulation often results in impaired protective immunity to vaccines, including malarial vaccine [16], BCG [17], hepatitis B virus (HBV) [18], and diphtheria toxin [19]. Moreover, findings in humans have also identified that helminth infections alter immune responses to measles, HBV, and Tetanus toxoid (TT) vaccines [10,20,21]. Of note, the use of anthelmintics, such as praziquantel (PZQ) or albendazole (ALB), significantly improves vaccination efficiency [18].

As our old friends of human coevolution, helminths and their metabolites and secretions interfere with immune responses to permit their persistence in hosts while dampening immune reactivity to bystander antigens in various immune disorders [12]. Some previous reviews have proved that the powerful immunomodulatory ability of helminth in hosts not only provides new avenues to treat immunological diseases such as allergies and autoimmunity, but also affects immunogenicity and efficacy of vaccines [12,13,22]. Given that helminth infections are among the most common infectious diseases in the world, evaluating vaccine efficiency affected by helminth infections may provide critical information for selecting optimal vaccination programs in regions with high parasite prevalence. The influence of helminth infections on vaccination has been previously reviewed, indicating that immune responses to vaccines are inhibited by the presence of helminth infections [12,13,22,23,24]. However, there is no systematic summary of the related immunological mechanism by which helminths affect vaccine effectiveness. This review aimed to provide up-to-date findings on the effects of helminth infections and their treatment on vaccination in humans and animals, highlighting the underlying immunological mechanisms. Moreover, the potential influence of helminth infections on SARS-CoV-2 vaccine was discussed, too. This review was based on literature searches in PubMed, NCBI, and related references from journal publications using mesh terms “Helminth”, “Parasite”, “Vaccine”, and “Efficacy”. Literature searches were conducted up to September 2022.

## 2. Effects of Helminth Infections on Vaccine Efficacy

Helminths are a wide variety of multicellular invertebrates with complex life cycles; among them, over 280 species can infect humans [25]. The main helminth species that cause human diseases include *Ascaris lumbricoides*, Hookworm, Whipworm, *Schistosoma*, etc. [4,26]. Generally, light to mild helminth infections usually do not cause obvious clinical symptoms, which delay treatment, and thus, lead to chronic infections. During long-term chronic infections, helminth-induced immune modulation not only reduces specific immune responses to the parasite itself, but also suppresses immune responses to other pathogens and antigens in hosts [27], making the hosts more susceptible to infection with other pathogens, such as human immunodeficiency virus (HIV) [28], *Mycobacterium tuberculosis* [29,30], malarial parasites [31], and even cancer [32].

### 2.1. Human Studies

Tuberculosis remains widespread among adolescents and adults, and adolescents are a target population for TB booster vaccines [33]. It is reported that the peak levels of helminth infections typically occur in hosts aged between 10 and 14 years in endemically infected communities [34]. Therefore, it is necessary to assess the impacts of helminth infections on TB vaccines in parasite endemic areas. One study in Uganda shows that *Schistosoma mansoni* infection does not affect the immunogenicity and safety of MVA85A, a model candidate tuberculosis vaccine, in BCG-vaccinated Ugandan adolescents [35]. Conversely, another group reports that individuals with *S. mansoni* infection may be at risk of a more rapid decline in antibody levels post HBV vaccine vaccination. Therefore, treating schistosomiasis with PZQ before immunizations is beneficial for vaccination [21]. Studies have also shown that *A. lumbricoides* infection causes an impaired immune response to oral cholera vaccines and a significantly lower level of vibriocidal antibodies, while ALB deworming treatment can restore the decreased levels of vibriocidal antibodies in infected children [36]. 

In tropical areas, vaccinations to prevent viral infections are often less effective than expected. It has been found that children infected with *S. mansoni* have much lower levels of anti-measles IgG antibodies than the healthy ones after measles vaccination in the Lake Victoria region of Uganda; even deworming treatment with PZQ cannot rescue the impaired long-term immune protection in children [10]. This finding is in line with the lower response to measles vaccination among children in rural Cameroon [20]. In clinical studies of another viral vaccine, poliovirus vaccine-specific IgG antibodies are decreased in children infected with *S. mansoni*, and the effect of *S. mansoni* infection on vaccination against poliovirus increases with age [37]. Furthermore, *S. mansoni* infection can impair the persistence of HBV vaccine-specific antibody responses, and the removal of parasites following PZQ treatment improves vaccine protection [21].

In addition, there is an association between helminth infections and other parasite vaccination. Compared with helminth-negative volunteers, the anti-GMZ2 IgG concentrations are significantly downregulated in the *Strongyloides stercoralis*-infected group and upregulated in the *Schistosoma haematobium*-infected group, after vaccination of malaria vaccine candidate GMZ2 against *Plasmodium falciparum* in children [38]. The Kremsner group found that children uninfected with *Trichuris trichiura* have a 3.4-fold higher antibody response to the GMZ2 than infected children in Gabon, along with moderately increased memory B-cell responses [39]. Both results indicate that it is necessary to account for helminth-mediated hypo-responsiveness of immune reactions during malaria vaccine development programs in the future.

In sum, all these results suggest that helminth infections affect immunogenicity and efficacy of bacterial, viral, and other parasitic vaccines in humans from endemic countries. Helminth infections can inhibit either the short-term immune response to vaccines or the long-term antibody titers after vaccination, and part of the inhibition effects can be rescued by deworming treatment (Table 1).

### 2.2. Animal Models

Experimental animals are ideal models to explore the effects of chronic helminth infections on vaccine efficacy and its detailed mechanism. Numerous animal studies have shown that chronic helminth infections reduce the effectiveness of various vaccines. Compared with the uninfected group, the colony forming units (CFU) of *M. tuberculosis* is significantly increased, accompanied by reduced air exchange area of the lungs after BCG inoculation in *S. mansoni*-infected mice, indicating that chronic *S. mansoni* infection significantly reduces the efficacy of BCG vaccination against *M. tuberculosis*
**[17]**. Apiwattanakul et al. assessed the potential role that the high burden of helminth infections in the countries targeted for vaccination may have on vaccine effectiveness in a mouse model. Chronic infection with *Taenia crassiceps* leads to impaired murine antibody responses to polysaccharide vaccines against *Streptococcus pneumoniae*. In addition, antibodies taken from *T. crassiceps*-infected, vaccinated mice do not protect against pneumococcal challenges in vitro and in vivo [43]. Interestingly, it is proposed that this immunosuppression caused by helminth infections can be improved by multiple vaccinations. Infection with *Nippostrongylus brasiliensis* leads to impaired immune responses to *Salmonella typhimurium* vaccine, evidenced by a decrease in IgG titers, while the IgG antibody titers can be recovered with the help of memory B cells by secondary immunization [44].

Another field that has been studied at the animal level is the impact of helminth infections on viral vaccines. A study of mice shows that infection with *S. mansoni* significantly reduces HIV-specific cellular responses. Furthermore, the cellular immune response is partially restored, but the humoral immune response cannot be restored by PZQ deworming treatment. Additional data suggest that the retaining of *S. mansoni* eggs contributes to a lack of full immune restoration after anthelminthic therapy in the mice [45]. A chronic *Schistosoma japonicum* infection can inhibit the immune responses to HBV vaccine and lead to lower production of HBV-specific antibodies in HBV-vaccinated mice [18]. In the primate olive baboon (*Papio anubis*) model, chronic *S. mansoni* infection elicits significantly reduced levels of HPV specific IgG antibodies, suggesting a compromised effect on the HPV vaccine. However, deworming treatment with PZQ before vaccination can restore this negative impact of *S. mansoni* infection [46]. A more recent study has shown that inoculation efficacy of the human pathogenic 2009 H1N1 influenza virus vaccine is severely impaired in acute or chronic *Litomosoides sigmodontis*-infected mice. Concurrent helminth infection results in a 10-fold reduction in the neutralization capacity of serum hemagglutinin (HA)-specific antiviral IgG in vitro [47]. The results obtained from infected mice with *L. sigmodontis* as a model for chronic human filarial infections also confirm that helminth-infected mice produce fewer neutralizing influenza-specific antibodies than vaccinated naïve control mice [48]. Both these data suggest that basic influenza vaccination regimen is not sufficient to confer sterile immunity in the context of helminth-induced immunosuppression, highlighting the risk of failed vaccinations in helminth-endemic areas. However, it is encouraging that a combination of deworming and prime-boost vaccination regimen can restore the efficacy of vaccination against influenza in helminth-infected mice, although applying this improved prime-boost regimen did not restore protection in untreated *L. sigmodontis*-infected mice [49].

The influence of parasite infections on vaccines for other infectious diseases has also been demonstrated in animal models. It is reported that both cattle and mice infected with *Toxocara*
*canis* have reduced antibody levels to Bovine herpesvirus type 5 (BoHV-5) vaccine in comparison with the uninfected animals [50]. Chronic *L. sigmodontis* infection significantly inhibits vaccination efficacy against *Plasmodium berghei*, while the efficacy can be restored by a more potent vaccine regime with recombinant live *Salmonella berghei*-expressing *P. berghei* proteins [51].

Although these aforementioned studies vary in helminth species, vaccine types, and animal models, all the results suggest that helminth infections may compromise vaccine efficacy, which is consistent with the findings obtained from human studies (Table 2). In most cases, the use of anthelmintic agents or the replacement of a more effective vaccine regime can at least partially restore the inhibition caused by helminths, and thus, improve vaccination effectiveness. Therefore, to improve the efficiency of vaccination in helminth-endemic areas, it is necessary to further explore the molecular mechanism by which helminth infections regulate immune responses post vaccination, which will provide strategies for improving the efficiency of vaccination in parasite endemic areas with high prevalence.

## 3. Immunological Mechanisms of Helminth Infections Affecting Vaccination Efficacy

Helminths and Humans share long co-evolutionary histories, which can be found in the earliest writings and have been confirmed by the finding of parasites in archaeological material [54]. As our old friends of human coevolution, helminths and helminth-derived molecules interfere with immune responses to permit their persistence in hosts, while dampening immune responses to allergens and autoantigens in various immune disorders [55,56]. There are several sophisticated mechanisms by which helminths manipulate host immune responses (Table 1 and Table 2). 

### 3.1. Immune Shift Caused by Helminth Infections

Helminths are known as the strongest natural inducers of type 2 immune responses, characterized by significant proliferation of Innate lymphoid cell type 2 (ILC2) and T helper 2 cells (Th2) that secrete cytokines such as interleukin-4 (IL-4), IL-5, and IL-13 [57,58]. Most vaccines provide effective protection by inducing Th1 responses in hosts. In general, Th1 and Th2 responses are antagonistic to each other; therefore, the Th2 polarization induced by helminth infections can affect the efficiency of vaccine administration.

Our previous study has shown that *S. japonicum* infection results in a Th2-dominated immune state (evidenced by higher IL-4 and IL-5 concentrations) that inhibits the production of anti-HBs antibodies and Th1 cytokines (IFN-γ and IL-2). Once Th1/Th2 cytokines are restored to an immune equilibrium by deworming, anti-HBs antibody levels return to normal [18]. Another study reports that *S. mansoni* infection reduces the protective efficacy of BCG vaccination against *M. tuberculosis* by polarizing the general immune responses to a Th2 profile in a mouse model [17]. IgG2a and IgG2b subtype antibodies are two indicators of Th1 type immune responses, while IgG1 represents a Th2 type immune responses. It has been shown that *S. mansoni*-infected mice have impaired responses to candidate HIV vaccines, which is associated with a suppression of IgG2a and IgG2b, and a significant reduction in the IFN-γ/IL-4 ratio [45]. Chronic *L. sigmodontis* infection significantly interferes with vaccination efficacy of malarial vaccine via decreasing the number of antimalarial-specific CD8 T cells and production of vaccine-specific IFN-γ and TNF-α in BALB/c mice [51].

These observations are also verified in human studies. Nutman et al. report that vaccination efficacy with the live oral cholera vaccine CVD 103-HgR is associated with a Th1 cytokine response (IL-2 and IFN-γ) to cholera toxin B subunit (CT-B), while infection with *A. lumbricoides* diminishes the magnitude of this response and ALB treatment before vaccination can partially reverse the deficit in IL-2 [59]. Consistent with the ‘old friends’ hypotheses, *A. lumbricoides* infection is associated with dampened Th1 cytokine immune responses to in vitro stimulation with H1N1 and lipopolysaccharide (LPS) antigens in Tsimane forager-horticulturalists in the Bolivian Amazon [40]. In addition, *S. mansoni* infection results in reduced protective efficacy against hepatitis B and tetanus vaccines in adults in western Kenya, and increased IL-5 concentration is observed in vitro cultures of peripheral blood mononuclear cells (PBMC) [21]. Moreover, one study shows that deworming treatment leads to increased cell proliferation responses in PBMC and decreased concentration of IFN-γ in the cell culture supernatant, suggesting that deworming treatment may reverse the state of Th2 immune shift in hosts [9].

Taken together, these findings suggest that helminth infections induce a polarization toward Th2 in host immunity, which impairs the Th1 immune response required for vaccine effectiveness.

### 3.2. Immune Regulation of Helminths

In addition to inducing the polarization of host immune responses towards a Th2 type, helminths have evolved sophisticated mechanisms to regulate immune responses, including inducing regulatory T cells (Treg) in hosts. IL-10 and TGF-β, two functional cytokines secreted by Treg cells, have negative immuno-regulatory effects, which can inhibit effector T cell immune responses to various antigens [60,61].

The vaccination-induced specific IgG antibody subtypes IgG2b, IgG2c, and IgG1 against the human pathogenic 2009 pandemic H1N1 influenza A virus, are drastically impaired due to infection with *L. sigmodontis* in mice. Mechanistically, the suppression is associated with a systemic and sustained expansion of IL-10-secreting Treg cells and is partially abrogated by in vivo blockade of the IL-10 receptor [47]. *Heligmosomoides polygyrus* infection results in a strong IL-10 and TGF-β1 response, and thus, suppresses the protective efficacy of the vaccine against malaria challenge, indicated by the inhibited levels of malaria-specific IgG, IgG1, and IgG2a in mice [16].

Data from human studies support the results from the above animal models. Maria et al. investigate the influence of Treg activity on proliferation and cytokine responses to BCG and *P. falciparum*-parasitized RBC in Indonesian schoolchildren. The results demonstrate that geohelminth-associated Treg has a stronger suppressive activity and reduces the immune responses to bystander antigens of mycobacteria and plasmodia, although there are no changes in the frequency of Treg cells [41]. Another study in Ethiopia evaluates the impact of anti-helminthic therapy prior to BCG vaccination on the immunogenicity of BCG vaccination in helminth-infected populations. The data suggest that chronic helminth infection suppresses the human immunogenicity of BCG, manifested as a significant reduction in the levels of IL-2 and IFN-γ, which is associated with increased TGF-β production by Treg cells but not with enhanced Th2 immune response [42]. Furthermore, Malhotra and colleagues investigate how maternal prenatal infections and newborns’ antihelminth cytokine profiles relate to IgG responses to standard vaccination during infancy in Kenya. The findings suggest that antenatal sensitization to schistosomiasis or filariasis and related production of antihelminth IL-10 at birth are associated with reduced anti-vaccine IgG levels in infancy, with possibly impaired immune protection against diphtheria toxoid (DT), HBV, and Haemophilus influenzae b (Hib) [62]. 

Collectively, helminth-induced immune modulation may have important consequences for vaccine trials in endemic areas, which is closely related to the frequency and functional activity of Treg cells in hosts. 

### 3.3. Other Related Immunological Mechanisms

Although lots of studies have shown that helminth infections suppress host immune responses to vaccines through immune polarization and immune regulation, some researchers have pointed out other potential mechanisms by which helminth infections lead to altered vaccine immune responses. Haben and colleagues reported that *L. sigmodontis* infection downregulates the number and frequency of vaccine-induced follicular B helper T cells, and thus, indirectly reduces the number of antigen-specific B cells as well as the Th2-associated IgG1 and Th1-associated IgG2 responses in mice [52]. Another study indicated that *T. crassiceps* infection reduces the proliferation of B cells in the spleen germinal center and, thereby, impairs the vaccine-induced protection against pneumococcal challenge in a mouse model [43]. A recent work reports that chronic *S. mansoni* infection impairs the sustainability of vaccine specific antibody responses in poliovirus-vaccinated children living in endemic areas and infected mice, which is associated with lowering the survival of plasmablasts and plasma cells by schistosomiasis in the bone marrow (BM). The good news is that PZQ treatment ameliorates the negative influences of the parasite on vaccine immunity, through improving BM plasma cell responses and partially restoring long-term specific serologic memory responses in both *S. mansoni*-infected children and mice [37]. Moreover, *H. polygyrus* infected mice have reduced T and B cells in superficial skin-draining lymph nodes (LNs), resulting in LNs atrophy and altered lymphocyte composition. The generalized reduction in the lymphocyte pool leads to reduced peripheral responses to BCG vaccination, while anti-helminthic treatment mends responses to BCG by regaining full-scale LN cellularity [53].

In summary, these findings suggest that helminths have complex and diverse mechanisms to affect immune responses, and thus, cause suboptimal vaccine immunity maintenance in hosts.

## 4. Do helminth Infections Interfere with COVID-19 Vaccine Efficacy?

Severe acute respiratory syndrome coronavirus 2 (SARS-CoV-2) is a newly emerged virus that caused Corona Virus Disease 2019 (COVID-19), which is highly infectious and easily mutates [63,64]. According to the WHO statistics, as of August 19, 2022, there are more than 590 million confirmed cases and more than 6 million deaths worldwide [65]. 

It is worth noting that the relatively low impact of COVID-19 in tropical and subtropical regions of the world coincides with areas of highly prevalent helminth infections, according to data from global helminth endemic countries [66]. In their recent publication, Bradbury and colleagues reported the possible negative interactions between helminth infection and COVID-19 severity in helminth-endemic regions, which might be associated with polarizing Th2 immune responses and reduced hyperinflammation caused by helminths during the co-infection [67]. Conversely, it has been suggested that helminth infections may suppress an effective immune response to SARS-CoV-2 at the early stages of infection, thereby increasing the morbidity and mortality of COVID-19 [68]. Alternatively, Oyeyemi et al. reported that the prevalence of schistosomiasis is associated with negative COVID-19 outcomes, and higher PZQ treatment coverage can reduce COVID-19 active cases and improve the recovery rate in African countries [69]. Another two studies have proved that patients co-infected with parasites are associated with reduced COVID-19 cases or deaths in Ethiopia and Uganda, which suggests that parasite-driven immunomodulatory response might mute hyperinflammation in severe COVID-19 [70,71]. Conversely, other findings have highlighted that besides helminth infections, other factors, such as lack of extensive testing, BCG vaccination, and unreliable information of case fatality rates and cause of death, could be the possible reasons for low lethality in sub-Saharan Africa [72,73]. Therefore, more solid data and further research are required to investigate the effects of helminth infections on COVID-19 and their interaction in endemic areas.

Effective global vaccination is crucial to the resolution of the COVID-19 pandemic. As of 19 August 2022, more than 12 billion doses of vaccine have been administered throughout the world [65]. Previous studies of other vaccines discussed in the preceding text suggest cause for concern on effects of helminth infections on COVID-19 vaccination programs. There is growing evidence that vaccine efficiency is reduced where helminth infections are prevalent, with reduced efficiencies shown for BCG, tetanus, diphtheria, measles, and malaria vaccines [13]. It has been reported helminths reduce Th1- and Th17-induced antiviral activity and vaccination efficacy [74]. Based on findings of previous studies on helminth effects on vaccine efficacy of other respiratory viruses [43,47], it can be hypothesized that helminths might inhibit COVID-19 vaccine responses, especially in those people living in parasite endemic countries. Therefore, evaluating vaccine efficiency in helminth-infected populations may provide critical information for selecting COVID-19 optimal vaccines for use.

In addition, it has been reported that the first dose of a COVID-19 vaccine is attributed to non-immunoglobulin E (IgE)-mediated mast cell activation by vaccine excipients [75]. And helminth parasites have been known to induce type 2 immune responses characterized by an increase in IgE levels, eosinophilia, mastocytosis, and basophilia [76]. As such, it is reasonable to anticipate that people with helminth infections and type 2 immune responses might have a higher risk for both non-IgE-mediated and IgE-mediated anaphylaxis on receiving booster doses [71]. Moreover, using fractional doses of COVID-19 vaccines to cover a greater number of people is under consideration, due to current vaccine supply constraints in lower- and middle-income countries [77]. It is speculated that helminths might have the potential to reduce antibody titers against SARS-CoV-2 after vaccination with fractional doses, based on findings on helminth infections inhibiting other vaccines’ efficacy. 

Collectively, the immunomodulatory effects of helminths, both advantageous and disadvantageous, can decrease the severity of COVID-19 and COVID-19 vaccine efficacy, respectively [24]. Unfortunately, the effects of helminths on COVID-19 vaccine immunogenicity and safety remain unknown, since clinical trials of COVID-19 vaccines have not included a cohort infected with helminths [71]. Evidence-based data are urgently needed to identify the effects of helminth on COVID-19 vaccine efficacy in helminth-endemic countries, which will directly influence recommendations on whether deworming interventions are needed for COVID-19 patients in at-risk communities.

## 5. Conclusions

Growing findings in human studies, combining results from animal models, have confirmed that helminth infections alter the immune status of hosts, and therefore, suppress the host’s immune response to vaccination. Helminths, as our old friends, have a strong ability to modulate host immune responses through immune polarization, immune regulation, and affecting the development and function of lymphocytes. As such, there is a dire need for anthelmintic treatments and prophylactics to eliminate helminth suppressive impacts on vaccination. Given that helminth infections are among the most common infectious diseases in the world, evaluating vaccine efficiency in helminth-infected populations will provide critical information for selecting optimal vaccination programs, especially in light of the ongoing vaccination campaign to control the COVID-19 pandemic.

## Figures and Tables

**Table 1 pathogens-11-01163-t001:** Effects of helminth infections on vaccine effectiveness in human studies.

Helminth Species	Vaccine Types, Study Subjects and Country	Major Findings	Immunological Mechanisms	References (Year)
*Schistosoma mansoni* and *Schistosoma haematobium*	Measles vaccine; Ugandan and Cameroonian children	Decreased antibody production		[10] (2019); [20] (2018)
*Schistosoma mansoni*	Hepatitis B and Tetanus toxoid vaccines; Kenyan adults	Antibody production decreases during long-term protection	IL-5 in PBMC cell culture supernatant ↑	[21] (2016)
*Ascaris lumbricoides*	Live attenuated oral cholera vaccine CVD 103-HgR; Ecuadorian children	Decreased immunogenicity and can be restored by albendazole	IFN-γ and IL-2 in PBMC cell culture supernatant ↓	[36] (2000)
*Strongyloides stercoralis* and *Trichuris trichiura*	GMZ2 malaria vaccine; Gabonese adults and children	Decreased antibody production and immunogenicity		[38] (2021); [39] (2012)
*Ascaris lumbricoides*	H1N1 vaccine; Tsimané adults	Decreased immunogenicity	IL-1β and IL-2 in PBMC cell culture supernatant ↓	[40] (2021)
Geohelminth	BCG; Indonesian children	Decreased immunogenicity	T cell proliferation and IFN-γ concentration ↓; Treg activity ↑	[41] (2010)
Intestinal helminth	BCG; Ethiopian adults Ethiopian college students	Decreased immunogenicity and can be restored by albendazole	IFN-γ and IL-12 in PBMC cell culture supernatant↓; IL-4 and TGF-β in PBMC cell culture supernatant ↑	[9] (2001); [42] (2008)

**Table 2 pathogens-11-01163-t002:** Effects of helminth infections on vaccine effectiveness in animal models.

Helminth Species	Vaccine Types and Animals Involved	Major Findings	Immunological Mechanisms	References (Year)
*Heligmosomoides polygyrus*	*Plasmodium chabaudi* AS antigen; C57BL/6 and BALB/c mice	Decreased antibody production	The levels of IFN-γ in spleen lymphocyte culture supernatants ↓; The levels of IL-4, IL-13, IL-10, and TGF-β ↑	[16] (2006)
*Schistosoma mansoni*	BCG; C57BL mice	Decreased immunogenicity	The concentration of IFN-γ and NO secretion in the culture supernatant of spleen cells ↓; IL-4 and IL-5 in the culture supernatant of spleen cells ↑	[17] (2005)
*Schistosoma japonicum*	HBV vaccine; BALB/c mice	Decreased antibody production and can be improved by praziquantel treatment	The concentrations of IFN-γ and IL-2 and mRNA levels ↓; The concentrations of IL-4 and IL-5 and mRNA levels ↑	[18] (2012)
*Schistosoma mansoni*	Diphtheria toxoid vaccine; CF-1 mice	Decreased antibody production		[19] (1997)
*Schistosoma mansoni*	Hexavalent (DTPa-hepB-IPV-Hib) vaccine; BALB/c mice	Antibody production decreases during long-term protection, which can be restored with praziquantel	B cell numbers and death of bone marrow plasmablasts and plasma cells ↓	[37] (2022)
*Taenia crassiceps* *Nippostrongylus brasiliensis*	Pneumococcal vaccine PCV13 and PPV23; Porin proteins OmpC, D, and F immunogen; BALB/c mice	Decreased antibody production	B cell numbers in the germinal center of the spleen ↓	[43] (2014); [44] (2014)
*Schistosoma mansoni*	HIV vaccine (SAAVI DNA-C2 (DNA), SAAVI MVA-C (MVA), HIV-1 gp140 Env protein); BALB/c mice	During long-term protection, antibody production decreases, and praziquantel partially restores cellular but not humoral immune responses	The ratio of IFN-γ/IL-4 in the supernatant of spleen cell cultures ↓, IL-10 levels ↑; The levels of IFN-γ and IL-2 ↓; The frequency of cytokine CD4+ and CD8+ T cells production ↓	[45] (2018)
*Schistosoma mansoni*	HPV vaccine; Subadult Papio anubis	Decreased antibody production		[46] (2019)
*Litomosoides sigmodontis*	Influenza season vaccine; C57BL/6 and BALB/c mice	Decreased antibody production	IL-10+ CD49b+ LAG3+ Tr1 cells and IL-10 production ↑	[47] (2019)
*Toxocara canis*	Inactivated BoHV–5 strain SV507/99; BALB/c mice and Cattle	Decreased antibody production	The mRNA levels of IL-12, IL-17 and IL-23 in spleen cells ↓	[50] (2020)
*Litomosoides sigmodontis*	ACT-CSP toxoid construct and the live bacterial oral vaccine; BALB/c mice and Wistar rats and cotton rats	Decreased immunogenicity	The number and function of CD8+ T cells ↓ IFN-γ and TNF-α production ↓	[51] (2012)
*Litomosoides sigmodontis*	DNP-KLH TD antigen; BALB/c mice	Decreased antibody production	The number of B cells and Tfh cells ↓	[52] (2014)
*Heligmosomides polygyrus*	BCG; C57BL/6 mice	Immunogenicity decreased	Peripheral lymph node lymphocytes ↓	[53] (2018)

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
