# Peer review of "A New Role for Old Friends: Effects of Helminth Infections on Vaccine Efficacy"

_pathogens, 2022, doi:10.3390/pathogens11101163_

Round 1
Reviewer 1 Report
The manuscript here by Feifan Zhu and colleagues presents a review of literature reporting the effects of concurrent helminth infections on vaccination efficacy. The authors report on a number of articles that indicate that helminth infections reduce the efficacy of vaccines against Measles, Hepatitis B, Tetanus/diptheria/pertusis, Cholera, Malaria, influenza, tuberculosis, and etc.,. The review includes studies in both humans and animal models, is well organized in content, and includes relevant and up to date studies/references. There is some slight English editing that should be performed, as the grammar on certain sentences is incorrect. though not so much as to interfere with understanding.
Reviewer 2 Report
A number of reviews systematically summarized literature examining the effect of helminth infection on immune response and vaccine efficacy. See below. So, this narrative review paper lacks novelty. The authors should justify the rationale for this review in the ‘introduction’ part. Limitations in the previous reviews if any and what this review adds to the knowledge should be discussed in detail. Otherwise, it would be a duplication of efforts. And some of these relevant published review papers are not cited in the current paper. In addition, the methodology for searching and screening of the literature isn’t clear.
Wait LF, Dobson AP, Graham AL. Do parasite infections interfere with immunisation? A review and meta-analysis. Vaccine. 2020;38(35):5582-5590.
Natukunda A, Zirimenya L, Nassuuna J, Nkurunungi G, Cose S, Elliott AM, Webb EL. The effect of helminth infection on vaccine responses in humans and animal models: A systematic review and meta-analysis. Parasite Immunol. 2022;44(9):e12939.
McSorley HJ, Maizels RM. Helminth infections and host immune regulation. Clin Microbiol Rev. 2012 Oct;25(4):585-608.
Vicky Gent, Simeon Mogaka, Effect of Helminth Infections on the Immunogenicity and Efficacy of Vaccines: A Classical Review, American Journal of Biomedical and Life Sciences. Volume 6, Issue 6, December 2018 , pp. 113-117. doi: 10.11648/j.ajbls.20180606.11
Akelew Y, Andualem H, Ebrahim E, Atnaf A, Hailemichael W. Immunomodulation of COVID-19 severity by helminth co-infection: Implications for COVID-19 vaccine efficacy. Immun Inflamm Dis. 2022 Mar;10(3):e573.
Table 1 is hard to read. Perhaps, a space or horizontal line separating study findings for each parasite spp. could help. A space or horizontal line separating study findings for each parasite spp. could also help improve the readability of Table 2.
Reviewer 3 Report
This is a very comprehensive review of the interactions between helminth infection and vaccine efficacy, which I think will be of value to researchers in the field.
As I have indicated, there are several errors of English in the text which should be addressed.
Also, on Line 167, the authors mention infection with Trypanosoma evansi - this is, of course, a protozoan.
Round 2
Reviewer 2 Report
The authors have not addressed my comments on the novelty and method of how this review was conducted. The introduction should clarify how this review is different from previous similar review articles on this topic and the new knowledge it adds to the subject. And the authors should also describe the literature searching and screening strategy in the paper.
Author Response
Response to the reviewer
The authors have not addressed my comments on the novelty and method of how this review was conducted. The introduction should clarify how this review is different from previous similar review articles on this topic and the new knowledge it adds to the subject. And the authors should also describe the literature searching and screening strategy in the paper.
Reply: Thank the reviewer for the critical comments. We have added the following information about the novelty of our review and the literature searching strategy in the revised manuscript.
L68-75: The influence of helminth infections on vaccination has been previously reviewed, indicating that immune responses to vaccines are inhibited by the presence of helminth infections. However, there is no systematic summary of the related immunological mechanism by which helminths affect vaccine effectiveness. This review aimed to provide up-to-date findings on the effects of helminth infections and their treatment on vaccine responses in humans and animals, highlighting the underlying immunological mechanisms.
L77-79: Literature searches in the review were conducted in PubMed, NCBI, and related references from journal publications. Our search contained the following keywords plus their variants: “Helminth”, “Parasite”, “Vaccine”, and “Efficacy”.